# Tele-Monitoring System for Chronic Diseases Management: Requirements and Architecture

**DOI:** 10.3390/ijerph18147459

**Published:** 2021-07-13

**Authors:** Lorenzo Mucchi, Sara Jayousi, Antonio Gant, Elisabetta Paoletti, Paolo Zoppi

**Affiliations:** 1Department of Information Engineering, University of Florence, 50139 Florence, Italy; sara.jayousi@pin.unifi.it; 2Nursing and Midwifery Department, Local Health Unit Toscana Centro, 50122 Florence, Italy; antonio.gant@uslcentro.toscana.it (A.G.); elisabetta.paoletti@uslcentro.toscana.it (E.P.); paolo.zoppi@uslcentro.toscana.it (P.Z.)

**Keywords:** eHealth, tele-monitoring, wireless body area network, security, quality of service, quality of experience

## Abstract

In the last years a large variety of eHealth services and Apps for professional medical users have been developed for different scenarios. The increasing elderly population (+100% in 2050) makes urgent to implement tele-medicine paradigm in the healthcare structures. The need of monitoring large number of patients distributed over the territory, together with the lack of medical resources, makes the adoption of Information Communication Technologies (ICT) crucial for the future healthcare services. This paper presents an ICT architecture model for the provision of tele-monitoring services within a novel proposed remote monitoring concept for healthcare, considering the new Family and Community Nurse (FCN). An integrated and personalized tele-monitoring solution is presented, through a detailed description of the reference network architecture and service platform. Moreover, the preliminary results of the experimental activities carried out for the evaluation of the system in terms of usability in operational scenarios are provided.

## 1. Introduction

The effects of the aging of the general population on the Health and Welfare system are considerable: the elderly gradual increase and the improvements in the patient’s life expectancy, thanks to the effectiveness of diagnostic and therapeutic pathways, lead to continuous increase of chronic patients. User-centered mobile health tracking apps are commonly used in healthcare and they have gained widespread popularity facilitating the maintenance of health and management of chronic conditions, allowing the sharing of clinical information over distances. Focusing on the management of patients with chronic diseases (diabetics, heart and respiratory diseases, etc.), the definition of a patient tele-monitoring system to support the Healthcare system is crucial to improve the quality of the healthcare.

### 1.1. Background on Statistics on Chronic Diseases and Aging

In 2015, the WHO presented the “World Report on Aging and Health” which shows that, worldwide, the improved prospects for life in the last 50 years have led to a percentage increase of over-64s and that, by 2020, there will be more than 60 year-olds than children under five. Italy ranks second in the world for longevity, after Japan [1]. Italy is therefore among the countries with the highest percentage of elderly people: currently, the life expectancy at birth of males amounts to 80.1 years, for women is 84.7 years old (ISTAT 2019). In Tuscany people over 65 years of age are about 25% of the population, of which 57% are women. The migration phenomenon only partially succeeds in slowing down the dynamics of progressive aging of the population. The effects of the general aging of the population on the Health and Welfare system are considerable: the gradual increase of the elderly and the improvements obtained in the life expectancy of a patient, thanks to the effectiveness of diagnostic and therapeutic pathways, make the number of chronic patients in the population tends over time to increase. Chronic-degenerative pathologies are more frequent in older age groups: in the 55–59 age group 53% suffer, and among the over 75-year-old the share reaches 85.3%. These are the most common chronic diseases: hypertension (17.4%), arthrosis/arthritis (15.9%), allergic diseases (10.7%), osteoporosis (7.6%), chronic bronchitis and bronchial asthma (5.8%), diabetes (5.3%). Excepting allergic diseases, all chronic diseases reported increase with age and with gender differences, generally with a disadvantage for women (ISTAT 2019). The chronic patients in Tuscany are almost one and a half million (ISTAT estimates from 2017 Multiscopo Survey), equal to 40.1% of the population and substantially in line with the Italian average (39.9%). The most frequent pathologies in Tuscany, among those surveyed, are arthrosis/arthritis and hypertension, respectively 16% and 15.7%. Given the increasing number of people who have more chronic diseases than those who only have one, the most frequent chronic condition is comorbidity (20.3%); it is now necessary to approach this issue with a different perspective, more oriented to overall clinical complexity of the person, rather than to the management of individual pathologies. Compared to 2018, there are about 20 thousand more chronic patients and this translates into a greater commitment, especially by the local services, in the daily management of the disease [2].

### 1.2. Background on Existing Tele-Monitoring Solutions

User-centered mobile health tracking apps are commonly used in healthcare and they have gained widespread popularity by facilitating the maintenance of health and management of chronic conditions [3,4] and by empowering individuals to contribute to their own well-being and health [5]. Catalysts of the mobile health Apps growth are the mobile technology availability, the increased affordability and access and the inherent convenience of devices. The WHO Global Observatory for eHealth (electronic Health) under the direction of Misha Kay, conducted three global surveys about the global spread of eHealth (in 2006, 2011 and 2016). The aim of the third global survey was to explore developments in eHealth since 2010 and the role eHealth plays in achieving Universal health coverage (UHC). It has become increasingly clear that UHC cannot be achieved without the support of eHealth. A large number of countries reported at least one mHealth (mobile Health) initiative (83%). The survey results clearly demonstrate that there has been impressive growth in almost all studied areas since 2010, reflecting country attitudes to eHealth now being considered fundamental to health system strengthening and innovation.

Provided examples of mHealth applications cover a broad spectrum from telephone helplines and text message appointment reminders to mobile tele-health and mobile access to electronic patient information. However, the process of embedding eHealth everywhere still has a long way to go, in terms of coverage and functionality. The picture now is quite mixed, with progress reflecting different national or local priorities [6]. To increase health App use efficacy, user-centered mobile health tracking Apps shall be easy to use. Health App use efficacy directly relates to the balance between ease of use and user burden.These Apps are typically commercially available through App distribution platforms (Google Play, Apple Store) and are commonly downloaded onto smartphones or tablet devices to provide a myriad of functions based on individual health care needs and patient conditions [7].

A literature search has attempted to identify the panorama of existing health Apps. This review revealed a very broad panorama of different health Apps in terms of target audience, main functions, costs and health care topics [8]. There is a wide variety of Apps with specific disease objectives and specific functions: Apps for diabetics, Apps for monitoring blood pressure [9], for patients with dermatological problems, Apps for mental health or surgery, Apps that help in taking therapy [10], App for cancer patients subjected to chemotherapy [11], App in hearing health care. The WHO 2015 survey also shows the percentage of countries reporting mHealth programmes for proving Health services (Figure 1).

Most studies analyze the offer of existing Apps on Android, Apple and Google App systems, creating App databases [12] or evaluating whether they are user-friendly and useful for the disease or target behavior [13]. Another problem that emerges is that, while the number of available Apps has grown substantially, there is no clear strategy on how providers should evaluate and recommend such Apps to patients [14]. Also, the WHO survey shows that, despite the rapid growth, very few Member States reported evaluations of government-sponsored mHealth programmes, thereby limiting knowledge of what works well and what mistakes to avoid [6]. Major stakeholders, including professional medical companies, insurers and policy makers, have largely avoided formally recommending Apps, which forces patients to get recommendations from other sources. In fact, some studies aim to help interested parties to overcome the obstacles to the review and recommendation of Apps, evaluating health care Apps for patients to manage their health appreciated by consumers.

### 1.3. Our Contribution

Analysing the statistics on chronic diseases and ageing and the existing ICT solutions for healthcare, some weak points that need to be addressed are identified:-the large number of existing mobile App or systems are focused on the treatment/management of a single pathology, while most patients are affected by multiple pathologies;-the correlation among different parameters from different sources (including environmental data) and the automatic sharing of the collected data among the monitored patient and the medical staff (e.g., nurses, general practitioners) involved in the management of his/her health is still missing; although some systems are patient-centered, they do not consider the patient as part of his/her healthcare path, which is crucial for improving his/her health status;-a real-time monitoring including data sharing and procedures for the validation of the acquired data operated by medical professional staff is needed.

The objective of this paper is to provide an ICT architecture model for the provision of tele-monitoring services within a novel proposed tele-monitoring concept for healthcare, including data gathering from multiple sources and multi-objectives analysis. Moreover, details of the developed components of the proposed architecture are presented together with some preliminary results of the validation of specific system functionalities.

We focus on the management of chronic diseases, considering patient’s co-morbidities and identifying the main professional users’ requirements in the management of an increasing number of patients. The increasing number of global population and the increase of the living age makes the healthcare of the future not sustainable without the introduction of tele-monitoring systems. It is worth highlighting that the proposed integrated and personalized real-time tele-monitoring solution is the result of the collaboration among engineers and medical staff; therefore, the design is focused on the real operative conditions and user requirements, including data validation procedures and alert management. This multidisciplinary co-design approach is the key element of the proposed system.

The paper is organized as follows: Section 1 gives an overview of statistics on chronic diseases and ageing and existing tele-monitoring solutions; in Section 2 a novel tele-monitoring paradigm for healthcare is presented, highlighting the healthcare cycle and the use case and requirements for the management of chronic diseases; the proposed ICT architecture model is described in Section 3, while Section 4 reports details on the user interface for the interaction with the system platform. In Section 5 the experimental activities are described and in Section 6 the preliminary achieved results are discussed. Finally, conclusions are drawn in Section 7.

## 2. A Novel Tele-Monitoring Concept for Healthcare

The objective of this section is to provide an overview of the healthcare context, including some examples of existing tele-monitoring solutions. Starting from this analysis and focusing on the management of chronic diseases, the use case and the end-user requirements for the proposed system are discussed, with the aim to introduce the ICT architecture model.

### 2.1. The Healthcare Cycle

In the last few years a large variety of eHealth services and smartphone applications for professional medical users have been developed for different situation contexts (well-being, drugs reference and management, training, medical news and information sharing). Most of the provided services and systems aim at supporting medical communities in their activities including critical situation contexts. In the Health Cycle (Figure 2) the four main areas in which eHealth can be used are represented.

One of the goals of health is to ensure the well-being of its citizens. Advanced ICT solutions for preventive care, activity monitoring, health and wellness education and sports have been defined by combining ambient and wearable sensors [15]. eHealth can also be used for health investigations with the aim to achieve the appropriateness of the primary care system, the rationalization of healthcare expenditure (reduction of improper access to the hospital and the rehabilitation in the emergency room, the reduction of expenditure on pharmacological, specialist and diagnostic therapies, etc.), the reduction of complications and disability with consequent improvement in the quality of life of people. Finally, the adoption of the proposed solution in the Healthcare System is expected to contain the healthcare spending in terms of reduction of improper access to hospital/emergency room and reduction in mortality and disability resulting from poor adherence to the prescribed therapy.

The use of telemedicine has emerged as a winning strategy for reducing re-admissions, improving patient health and reducing healthcare costs. In [16], the authors consider many small studies showing a reduction in all-cause mortality by 34% and hospitalizations for heart failure (HF) by 23% due to the adoption of telemedicine. A retrospective case-control study demonstrates a reduction in total hospital days, overall hospitalizations and length of stay without increasing outpatient visits and urgent care with the support of home tele-monitoring integrated into the healthcare system [4]

Telehealth can also significantly improve healthcare provision in rural and remote emergency departments through the development of cost-effective models that remain similar in quality to physician-staffed services. Using two-way interactive technology and telecommunication through tele-health improves collaboration through telephone or videoconferencing consultations between referring hospitals and receiving hospitals which may reduce secondary overtriage and optimise patient management within community hospitals [17]. In critical presentations, telehealth has reduced morbidity and mortality rates, hospital admission time and cost of patient care [18]. The use of mHealth and eHealth is also useful in the surgical field. The length of in-hospital stay after general surgical and gynecological procedures has decreased significantly due to a growing trend in day-care surgery, introduction of minimal invasive techniques, and enhanced recovery after surgery programs. Electronic health (eHealth) can be a suitable tool to optimize the quality of perioperative care of patients who will undergo general surgical and gynecological procedures. eHealth can provide tailored information, increase patients’ self-management, and has interactive communication features [19]. Furthermore, it has the potential to empower patients, to motivate patients, and turn them into more active and effective managers of their own health. The need of monitoring an increasing number of patients distributed over the territory, satisfying patient assistance requirements and coping with the lack of medical resources make the adoption of ICT crucial for the future healthcare services.

Focusing on the management of patients with chronic diseases (heart disease, diabetics, people with respiratory diseases, etc.), the definition of a patient tele-monitoring system to support the Healthcare system is of fundamental importance for improving the quality of the healthcare system. The management of chronic patients at home aims at empowering the patient for a proper self-care and self-monitoring, as well as direct professional assistance when necessary. A good compliance in some parametric/anthropometric checks and the prescribed therapy are essential elements for keeping a good health status, the quality of life, the reduction of hospital admissions, the use of diagnostics for images and laboratory, with an indirect impact on pharmaceutical, diagnostics and specialist spending. The integration and correlation of data (biometric parameters, physical activity, adherence to therapy) through advanced algorithms allow an effective monitoring of the patient’s health status, providing an immediate overview of the progress of the made measurements and a quick identification of any anomalies found, activating automatic alert for patient prevention/management actions.

### 2.2. Management of Chronic Diseases

#### 2.2.1. Use Case

The healthcare system need of monitoring an increasing number of patients distributed over the territory and the lack of medical resources to cope with the patient assistance requirements make the adoption of ICT crucial for the future healthcare services. Focusing on the health status monitoring of patient affected by chronic diseases, the possibility of executing some basic exams to assess the patient general status and automatically share the acquired clinical data with all the members of the medical team that follow him/her is of fundamental importance for the management of the patient health. The periodic measurement of some patient’s biometric parameters, the monitoring of his/her daily physical activity and therapy compliance, which represent the main inputs for a follow up program and the evaluation of the patient health status, cannot be performed efficiently without the use of technologies. The sharing of these data among the actors involved in the management of patient‘s health is not feasible without the definition of a telecommunication network for the data acquisition and transmission and a system platform for data storage, analysis and distribution. The use of a set of wireless wearable medical devices and sensors allows the periodic measurement of patient’s biometric parameters and the automatic recognition of his/her daily physical activity guaranteeing an efficient tele-monitoring service for patient regardless of his/her location. The data can be automatically gathered by the patient’s smartphone equipped with a customized App, acting as a “coach” for the patient (e.g., therapy assumption reminders). The real-time collection and the analysis of data related to the patient’s health status are extremely useful for the patient active monitoring. Preventive identification of potential health risk and consecutive planning of adequate actions, such as therapy calibration and management or additional exams booking, will benefit from the inclusion of innovative technological tools in the healthcare protocols and organization models. The storage of the collected information in the patient’s clinical health record and the access enabled to all the authorized people involved in his/her health management rely on the definition of a communication network architecture and interfaces able to support the secure data transmission and sharing. Moreover, the integration of such information with the ones collected by the several existing and non-integrated software programs adopted by the National Healthcare System for the management of patient health data need to be carefully addressed. The evaluation of the potential definition of interfaces among a new monitoring tools and the traditional ones is extremely important and aims at the integration of all the patient’s clinical data in a unique clinical health record regardless of the input sources. Based on the previous considerations, ICT solutions play a key role in the provision of an efficient and cost-effective patient tele-monitoring services and will support the medical professionals in their daily activities. ICT will ease and make feasible the assistance of patient at home, facing both the increasing number of patients that need to be assisted and included in a follow up program and the lack of adequate amount of medical resources.

#### 2.2.2. User Requirements

In [1] the FCN was introduced and addressed as the key contribution within the multi-disciplinary team of health professionals for the achievement of the 21 Health objectives of the XXI century. The FCN is in charge of helping people to adapt to their disease and chronic disability by assisting them and their family mostly at home. The goal is to keep under control and improve the health status of the family within the community, helping the family to avoid and manage health threats.

The four main characteristics of this new model are: proximity (providing for the identification of chronic patient); proactivity (recognizing early the latent needs of the population and driving them trough the clinical and social services network); equity and multiprofessionalism.

A simple and user friendly monitoring system to allow FCN to perform direct and remote patient monitoring is essential in this scenario. Real-time data sharing within the team in charge of the patient, together with the integration of data by the patient him/herself or by family is a key factor for the identification of any changes in the patient’s diseases and conditions and allows adequate and early actions to avoid worsening of clinical and health conditions.

## 3. ICT Architecture Model

This section aims at describing the proposed ICT reference network architecture. In particular, as a preliminary step, the system requirements are identified, then the architecture is presented and the different components are detailed. Finally the service platform capabilities are reported.

### 3.1. System Requirements

To meet the user requirements a modular telemedicine system to support medical staff in the provision of home healthcare assistance to patients affected by chronic diseases is proposed. An integrated and personalized tele-monitoring solution is defined for guaranteeing an efficient follow up of those patients that need to be assisted at home through a periodic check of some parameters (clinical parameters and habits).

The requirements for the proposed system derive from the analysis of end-user needs. In particular, the co-creation and co-design approach followed by the authors relies on their awareness that an effective ICT solution, which includes Human-Computer Interaction (especially in the healthcare sector) should be based on the adoption of a participatory design paradigm, that allows to satisfy the end-users’ needs and, consequently, improve their quality of experience. The participatory design (PD) main principles are deep engagement, interdisciplinary, individuality, and practicality. The proposed system comes from an interdisciplinary investigation based on the PD approach since the very beginning of the study. The deep collaboration between engineers and end-users (e.g., registered nurses, medical staff) has allowed the identification of the existing needs. This has enabled the definition of real use cases taken as a reference to show the benefit of the proposed system with respect to the already existing and sometimes adopted technologies. Moreover it has allowed the identification of the main potential challenges that need to be taken into account both in the development and validation phase, but also to make the system be accepted and therefore adopted by the end-users.

In detail, the end-user requirements for patient monitoring (self- and professional monitoring) have been identified, focusing on the main application context of the proposed solution: *Tele-monitoring services for patient assistance and follow up at home*. These requirements have been defined thanks to a deep collaboration between the Department of Information Engineering and the Local Health Unit Toscana Centro, in particular the Nursing and Midwifery Department. This collaboration has been fundamental for the design of the overall system and the development of the tested components of the proposed architecture. In detail, it allowed the definition of the system functionalities and interfaces, and consequently the service requirements, in compliance with the medical professional end-users needs, also taking into account their organizational models and operative contexts.

As a result of this collaboration, three main layers of patient monitoring have been identified for the proposed system. These are:-*Parameters Monitoring*, through periodical check of some parameters performed by the patient him/herself or by nurses.-*Physical Activity Monitoring*, through the analysis of patient’s movement during his/her daily life.-*Therapy Compliance Monitoring*, through a daily automatic interaction with the patient and feedback requests regarding the correct assumption of the prescribed therapy.

Moreover, the provision of an effective healthcare service relies also on the capability to address a large variety of scenarios. Four main features are identified for the definition of a particular scenario. These are:-*Mobility level*. It is low in indoor context(at work, home, room hospital), medium in outdoor pedestrian context or high in vehicular context.-*Network availability*. The scenario can be characterized by very limited connectivity or broadband connectivity or resilient broadband connectivity.-*Medical resources availability*. It can be characterized by: (i) absent/very limited (e.g., rural area/developing countries); (ii) medium number of medical professionals not covering all the medicine areas and medium medical diagnostic instruments availability (e.g., small Hospitals); high number of professionals of different medical specialization and advanced technological diagnostic instruments (e.g., high specialized Hospitals)-*Priority of intervention*. It is low in case of periodical check-up, medium in warning situations and high in emergency contexts.

### 3.2. Reference Network Architecture

Figure 2 depicts the proposed reference communication network architecture for the provision of eHealth tele-monitoring services, highlighting its four layers:*BODY-Layer*. It consists of a set of body sensors/devices, able to gather different patient’s parameters. In-body and on-body sensors are considered. The former are represented by tiny sensors, which can be implanted or swallowed, while the latter are devices worn by patient for bio-parameters measurements (e.g.,: blood pressure, pulse oximeter, ECG sensors) acquisition.*ENVIRONMENT-Layer*. It includes indoor and outdoor environmental sensors for information gathering (e.g., temperature, air quality, humidity) and user positioning (e.g., GPS, video-monitoring) identification, This allows to provide location-based and personalized services also based on the user life style.*EDGE-Layer*. It is responsible for short-term analysis of the acquired data and for user-system interactions (e.g., through a smartphone App). It allows the quick processing and identification of anomalous patient’s conditions and the consecutive warning notification.*CLOUD-Layer*. It is in charge of the eHealth Service and the long-term data analysis through the adoption of advanced algorithms for data cross-correlation.

To allow the collection, processing and transmission over Internet of patient health data, the three intra- and three inter- communications layers (ComLayer) can be identified. These are:-*Intra-Com-1*. It represents the communications within the human body (in-body), such as data exchange among nano-devices and/or implants, or biological communications, where molecules act as a mean for encoding, transmitting and receiving information (molecular communications).-*Intra-Com-2*. It consists of the communications between in-body and on-body devices. The latter may act as sink nodes for the collection of data gathered by in-body sensors (nano-devices or implants) and for delivering the acquired information to off-body devices [20].-*Intra-Com-3*. It represents an optional communication layer of the EDGE-Layer that may exist in case the gateway function is not integrated into the sensors data collector. Therefore it interfaces the data collector (HUB) to the WAN device (gateway).-*Inter-ComLayer-1*. It represents the communications between the BODY-Layer and the EDGE-Layer. It enables on-body devices or sensors to interface with a hub/gateway (e.g., a smartphone).-*Inter-ComLayer-2*. It represents the communications between the ENVIRONMENT-Layer and the EDGE-Layer. It enables environmental sensors to interface with a hub/gateway.-*Inter-ComLayer-3*. It consists of all the communications between the EDGE and the CLOUD-Layer, that allow the transmission of data from the gateway to the cloud (e.g., 3G/4G/5G, WiFi, SigFox, Low Power Wide Area Network).

It is worth highlighting that the authors from the Department of Information Engineering are part of the ETSI SmartBAN Working Group (WG) and actively participate in the WG activities. The proposed architecture is based on the high level concept of the architecture defined within the WG activities [21]. On the other hand, the healthcare cycle derives from: the revision of the state of the art on the healthcare available services, the analysis of the main user requirements and the author’s experience in the healthcare sector.

### 3.3. The Service Platform

Considering the reference architecture (Figure 2), a service platform for satisfying the user requirements for chronic diseases management is proposed. The adoption of ICT eases the monitoring process and data sharing, supporting the FCN to perform his activities. The Platform consists of three components:

#### 3.3.1. Patient’s Parameters Monitoring

Periodical check of some parameters (e.g.,: weight, blood pressure, SpO2, glycaemia, hurt-level) is carried out autonomously by the patient him/herself (self-monitoring) or by nurses (professional monitoring). In both cases, the collected data are automatically stored in the patient clinical record and shared with the medical actors involved in the patient’s health management. Traditional and new wireless wearable and non-wearable medical devices can be used for the manually or automatic acquisition of the parameters, respectively. While a common smartphone is used for collecting all the measured data and transmitting them to the Cloud Service Center.

#### 3.3.2. Patient’s Physical Activity Monitoring

The patient physical activity is transparently monitored to evaluate patient’s lifestyle in terms of sedentary behaviour or active life. Through wearable or smartphone sensors, data related to patient’s movements are collected and processed in real-time, enabling recognition, analysis and storage of his/her physical activity. This allow to achieve habits information helping the health status analysis and guiding the patient to avoid bad lifestyle choices.

#### 3.3.3. Patient’s Therapy Compliance Monitoring

Opportunely scheduled reminders are provided through a customized App, acting as a patient’s coach for his/her therapy management. It guides him/her to be compliant with the assigned therapy, which can be remotely updated by the general practitioner. The daily patient interaction with the app reminders and the acquisition and storage of therapy compliance data allow the correlation of such information with potential health problems, giving medical staff precious inputs for patient’s health status evaluation and management.

### 3.4. Security and Privacy

It is well known by anybody that the security and privacy issues are extremely important when we talk about tele-medicine. Health-related data are very sensitive, and the user must be aware anytime about where data is collected and who has access to it. This paper does not focus on this topic, for the sake of space, but anyway this paragraph wants to summarize the main characteristics of a good policy of security and privacy when a tele-medicine architecture is on. Concerning our experimental activity, we only relied on the application layer security protocols, and on the fact that the data where stored in a private server without public access.

From the point of view of the legal aspects, the operations on the citizen’s personal and health data necessary for the provision of tele-medicine fall within the processing of sensitive data carried out by electronic/digital tools, which are governed by the provisions of Legislative Decree 196/2003. The methods and solutions needed for ensure confidentiality, integrity and availability of data must, therefore, in any case be adopted in accordance with the security measures expressly provided for in Legislative Decree no. 196/2003.

Each data transmitted has a fundamental importance in the diagnostic/therapeutic approach to the user which requires, in terms of qualitative, confidentiality and security aspects, scrupulousness mandatory and indispensable in its handling. Health data can flow from patient to cloud and from cloud to others (health operators) by using different channels (often wireless), which not always can be assumed secure. A trusted and secure domain for the health data should be assured, so that there is no segment which has a lower security then others [22,23]. In addition to classical cryptography, modern security techniques should be added, such as physical-layer security and crypto-key extraction from health signals in order to provide additional level of security and authentication of personal devices extracting the health data. Access policy to user’s health data should also be addressed, by making the patient aware of where the data is and who is requesting access to it. Blockchain technology can also be used to trace the access log to data and for non-repudiation of actions ordered or made on the patient’s data.

## 4. User Interface for System Interaction

The objective of this section is to provide details on the real system functioning by describing the user interface for the interactions between the end-users and the Platform. In particular, the developed Application is presented highlighting its usage in operative practical contexts.

The interface between the end-users and the system is represented by a customized Android Application which has been designed and developed starting from the identified user requirements. The main App functionalities have been differentiated based on the the different end users. Two main profiles have been defined: SELF and PRO. The former is for patient self-monitoring and the latter is for professionals periodic monitoring and data analysis ( Figure 3).

While the SELF profile is be able to collect bio-parameters, kinematic data and therapy compliance information, the PRO profile allows the patients clinical data analysis, evaluation, alert generation and therapy management based on the collected information. To properly design and develop the custom modular App, which, enabling the collection of data from sensors and the transmission of a pre-processed version to the cloud repository, different aspects have been considered. These can be grouped into: *Technical aspects.* The App main functionalities are defined based on the identified solutions and the main building blocks for data acquisition, processing and transmission are designed to allow the smartphone to interface both to the measurement instruments and sensors and the telecommunication networks. *User-oriented aspects.* The design of the application user interface is carried out focusing on user-friendly solutions.

### 4.1. Application Functionalities

The main App functionalities of the SELF and PRO profile and the monitoring of patient’s bio-parameters are described in the following. Based on the end-user authorization the APP shows and enables the access to specific functions. The SELF profile allows:-Acquisition of bio-parameters through the automatic interfacing between advanced wireless medical devices and the smartphone;-Manually insertion of bio-parameters measured with traditional medical devices, the patient owns;-Identification of who perform the measurement and when the measurement is performed;-Automatic and patient transparent acquisition of kinematic data (directly from the smartphone sensors or from specific wearable sensors);-Visualization of the last patient measurements;-Pre-processing of the acquired data and transmission to the cloud repository;-Daily interaction with the patient for therapy compliance monitoring;-Visualization of the prescribed therapy.

The PRO profile allows:-Acquisition of bio-parameters through the automatic interfacing between advanced wireless medical devices and the smartphone;-Manually insertion of bio-parameters measured with traditional medical devices;-Identification of nurse performing the measurement and when the measurement is performed;-Visualization of all the patient clinical health record, including the performed measurements opportunely detailed;-Pre-processing of the acquired data for a pre-alert generation in case of anomalous values and transmission to the cloud repository;-Management of the patient therapy based on the collected information;-Patients’ clinical data analysis and evaluation;-Generation and management of alerts/warnings.

### 4.2. Operative Use Case Example and Relative Information Flow

In order to provide a practical example of the functioning of the App, in the following an operative use case example is described, highlighting the relative information flow. To simplify the description, we assume that: Marco is the patient, Giovanni is the FCN and Luca is the general practitioner. Marco is Luca’s patient and he uses the App for (self) monitoring his blood pressure and weight on daily basis. Recently he is gaining weight and his pressure is not as stable as before. Luca received some warnings notifications from the App regarding to Marco’s parameters. Due to some anomalies in the collected Marco’s values, he decided to visit him and plan some periodic Marco’s check up performed by Giovanni. For one month, Giovanni went to Marco’s house to perform some bio-metrics parameters measurements (mainly weight, blood pressure and glycemia) and to check his overall health status. This data are saved to Marco’s clinical health record in real-time thanks to the App, allowing Luca to be updated on Marco’s health and measurements values. Luca can also verify the validity of the recorded measurements thanks to the periodic check-ups that Giovanni performed. After two weeks Luca, based on Marco’s historical health data and the still existing anomalies in the collected data, which have been notified to him by the App warnings, decided to further investigate on Luca’s conditions and booked for him some examinations. All these actions are traced by the App thanks to the association of each check up with a flag status that, beside indicating the normal, warning or alert status, provides information regarding booked or performed medical professional visits or examinations.

### 4.3. App Graphical User Interface

The App has been designed following a modular approach that allow an easy integration of future functionalities. This flexible solution allows also a scalability in terms of user typology and specific application contexts. The main idea is to provide an App whose main functionalities can be activated based on the patients clinical and management needs. The App Graphical User Interface (GUI) is intuitive and user friendly. Some screenshots of the App developed and tested functionalities are reported in (Figure 4 and Figure 5).

## 5. Experimental Activities

The objective of this section is to describe the experimental activities carried out in the framework of this study.

### 5.1. Experimental Activities Plan

Among the different service platform components, the patient’s parameters monitoring performed by nurses was developed and validated in a real context. The service user interface is represented by a properly designed and developed custom modular App. The App enables the collection of data from sensors and the transmission of a pre-processed version to the cloud repository. These functionalities are designed considering technical and user-oriented aspects. Medical end-users, nurses, healthcare managers are involved to get preliminary feedback. In detail, the experimental activities consists of two main phases:-*First phase*. Preliminary validation of the developed App.-*Second phase*. Extended validation of the App involving a number of nurses and real patients.

In the first phase, a small number of nurses have been involved to evaluate the App. No real patients have been involved. The App functionalities have been tested by simulating the measurements executions on patients. The collection of feedback has been based on the results of a preliminary questionnaire for evaluating the usability of the proposed solution. On the other hand, the second phase of the experimental activities, whose Ethical Committee approval process is still ongoing, will involve the introduction of the App on an experimental basis in clinical practice, including both nurses and patients of a selected area of Local Health Unit Toscana Centro (Italy).

To achieve the objectives of the experimentation and the evaluation of the QoE (Quality of Experience), data related to the following indeces were collected:approval ratingindex of functionalityusability indexacceptability index

These indices were measured according to the MOS (Mean Opinion Score) scale and were the output of a questionnaire submitted to the nurses involved in the pilot phase.

It is worth highlighting that an ad-hoc questionnaire has been defined by the Dept. of Information Engineering of the University of Florence for a preliminary evaluation of the usability of the App and the service developed components. The analysis of the data obtained allows to improve the App.

### 5.2. First Phase of the Experimental Activities: Steps and Details

Focusing of the first phase of this pilot study experimental activities, the main pursued objective is to evaluate the service usability in operational contexts.

The steps that have been followed are depicted in Figure 6.

In detail, a group of 10 FCNs belonging to the city area (family and community nurses of the Florence city district) has been identified. This group is representative of the group of nurses to whom the application is intended to and that will be involved also in the second phase. The inclusion criteria for the FCN selection have been the following: (i) FNC who have completed at least the 6-month length of service and have an android smartphone; (ii) no indications have been given regarding IT skills, because one of the aim of this study is to develop an App as user-friendly as possible.

A training session has been organized with the promoters of the study and the developers of the app for presenting the study protocol and the functionalities of the app and its use. Then the app was installed on personal nurses’ mobile smartphones and some demonstrative examples were carried out. The insertion of some simulated vital parameters have been performed, with the aim to make the nurses understand the App functionalities and basic usage. During this training session, nurses’ doubts and questions have been answered. App promoters and developers provided indications for reporting any system bugs during the next 20 days of the App test. Then the usability test campaign started: during the testing period (20 days), the nurses were requested to simulate at least one patient’s checkup a day, inserting the different parameters included in the monitoring service and checking the patients historical data including the presence of alert or warning messages, notified by the App in case of parameters values anomalies.

After 20 days, the nurses completed an evaluation questionnaire properly defined with the aim to get quality of experience feedback regarding both the App functionalities, usability and interface. The results allowed the identification of the advantages and the needed system improvements for its use in real operational contexts.

### 5.3. Patient’s Monitoring Parameters

The pool of parameters included in the simulated patient record was chosen on the basis of both current good practices (for example pain measurement) and on hypothetical problems and care needs (in accordance with the most frequent clinical cases). The App is designed to be able to enter all the possible detectable parameters (example: daily / hourly diuresis, mews score, pain, heart rate/respiratory pressure, saturation, weight, etc). Nurses are able to customize the pool of parameters detected in “that” specific patient so that they can review and enter the most frequently detected parameters more quickly. An “nurse worry index” has also been inserted to give the nurses the opportunity to express their opinions regarding what they consider important both from a clinical point of view or objectively detected, based on their “clinical eye” (given by the professional experience) that goes beyond the simple insertion and evaluation of the parameters.

## 6. Results Analysis

The objective of this section is to provide an overview of the experimental activities results, highlighting the received feedback and suggestions together with the impact of the proposed system in terms of strenghts, weaknesses, opportunities e threats.

The presented results refer to the first pilot phase of the project, which aims at involving a small group of end users belonging to the community of nurses for validating the App based on the professional point of view pursuing a collaborative and co-creative approach. On the other hand, the next and second phase of the experimental activities will aim at validating the app in terms of usability based on the patients point of view. They will be involved for receiving feedback, that will also be evaluated together with the ones received from the nurses.

### 6.1. Results of the First Phase of the Experimental Campaign

The results collected during the first phase of the experimental activities are mainly oriented to assess the usability of the system. In detail, the evaluation consists of:-the end-user validation of the developed App;-the evaluation of the effectiveness of the proposed Service;-the preliminary evaluation of the social impact of the proposed system for future extensions at national and European level.

As described in the previous section, the end-user feedback comes from the analysis of the distributed questionnaire. Some of the main questions that were included are:Should the Healthcare System adopt information and telecommunication technologies aimed at monitoring some chronic diseases?How do you evaluate the features offered by the App in support of the nursing staff (patient search/checkup, parameter annotation, historical display, alert management)?How do you evaluate the overall usability of the App (e.g., graphic interface, intuitive use, navigability)?How do you rate the App (as a whole monitoring system) and the remote patient monitoring system?Do you think this tele-monitoring system/solution can be an effective tool to support healthcare professionals for remote monitoring and patient health management?

In details, the quality of experience results (Mean Opinion Score scale) of the investigated aspects are reported in Table 1.

The results show that the main tested functionalities of the App can effectively support the nursing staff and this study may represent a first step towards the patient’s tele-monitoring and his active involvement in the management of his state of health. The App functionalities extension (e.g., new measurable parameters to allow greater monitoring flexibility for different specialities, activation of communications and urgent notifications also to other healthcare operators) would allow an evolution and an enhancement of the tool in its operational purposes. This preliminary test campaign represents the first step, which provides some guidelines for the implementation of the whole system, designed and developed following a modular approach.

### 6.2. Additional Received Feedback

In addition to the questionnaire, the involved RNs were also asked to provide some comments and suggestions. The most significant ones are listed in the following:-the App should be a tool used by all the staff involved in the patient’s health management;-the current App may be suitable for autonomous and family (or caregiver) monitoring (daily measurements of bio-metric parameters);-the alerts notification should be improved especially in case of urgent situations; the integration with the protocols adopted in different real operating conditions should be considered;-it would be useful to be able to identify more quickly the type of the performed check-up (in terms of parameters);-the development of a web interface (accessible from a PC) could be useful;-interfacing with the current management information systems in use in the healthcare sector may be useful;-the possibility of interfacing with other healthcare professionals and the need for an immediate feedback from the doctor in the event of an alert should be considered.

These suggestions together with the feedback coming from the second phase of the experimental activities will be used to further update the developed system and provide recommendations and guidelines for the adoption of the proposed solution.

### 6.3. Recommendation and Guidelines Road-Map

The App design has followed the current guidelines for the implementation of the tele-monitoring systems indicated also by the WHO and by the national and regional health plans in addition to the indications provided by the scientific societies for chronic diseases.

However, at the end of second experimentation phase, guidelines for the application of this monitoring model will be defined and disseminated, initially at the regional level.

In detail, once the app will be updated based on the received feedback of the second phase of the experimental campaign, its usage can be further extended to the entire territory of the Local Health Unit Toscana Centro (which covers an area of 5000 square kilometers and 1,500,000 citizens, it has 13 hospitals, 220 territorial care centers and including both urban areas with high population density and sparsely populated rural and mountain areas). In fact, from this experience it will be possible to develop solid guidelines for the adoption of a new patient-nurse monitoring systems dedicated to chronic patients and for the diffusion throughout the Tuscany region.

### 6.4. System Impact and SWOT Analysis

Regarding the social impact, the scalability, portability, low cost and the “user friendly” approach are the key factors for the adoption of the system by the medical community and the healthcare system. Ensuring efficient patient care by reducing costs and exploiting the availability of distributed medical resources partially simplify the complex hospitalization assistance and management system and address one of the current challenges of healthcare: patient empowerment.

In order to give an overview of the main study impact, the SWOT (Strenghts, Weaknesses, Opportunities e Threats) analysis is provided (Figure 7), highlighting the strengths, weaknesses, opportunities, and threats related to this study.

## 7. Conclusions

A novel tele-monitoring concept for healthcare is presented. User and system requirements are carefully identified to define a system able to support medical staff and nurses in their operative conditions and to let the patient feel part of his/her healthcare pathway.

The results of a preliminary test campaign on patient’s parameters monitoring component show the effectiveness of the developed and tested component of the proposed system in supporting nursing staff in the management of chronic diseases.

## Figures and Tables

**Figure 1 ijerph-18-07459-f001:**
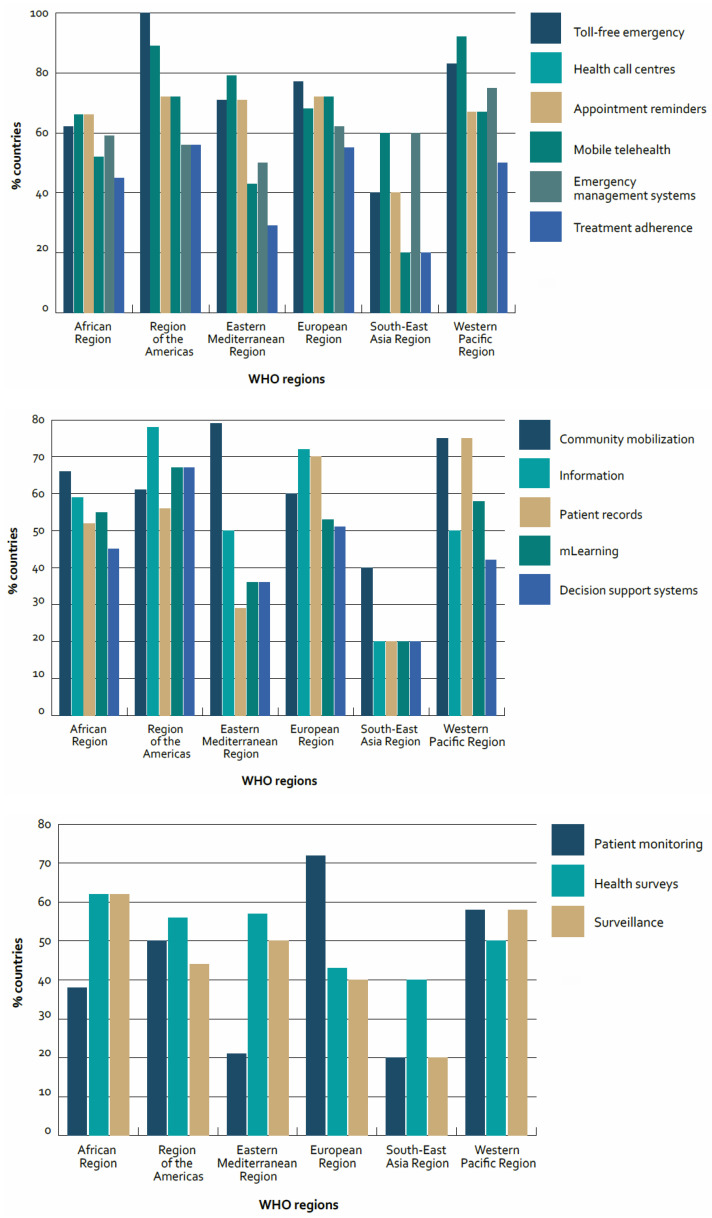
Percentage of countries reporting mHealth programmes for accessing/providing health services by WHO region [6].

**Figure 2 ijerph-18-07459-f002:**
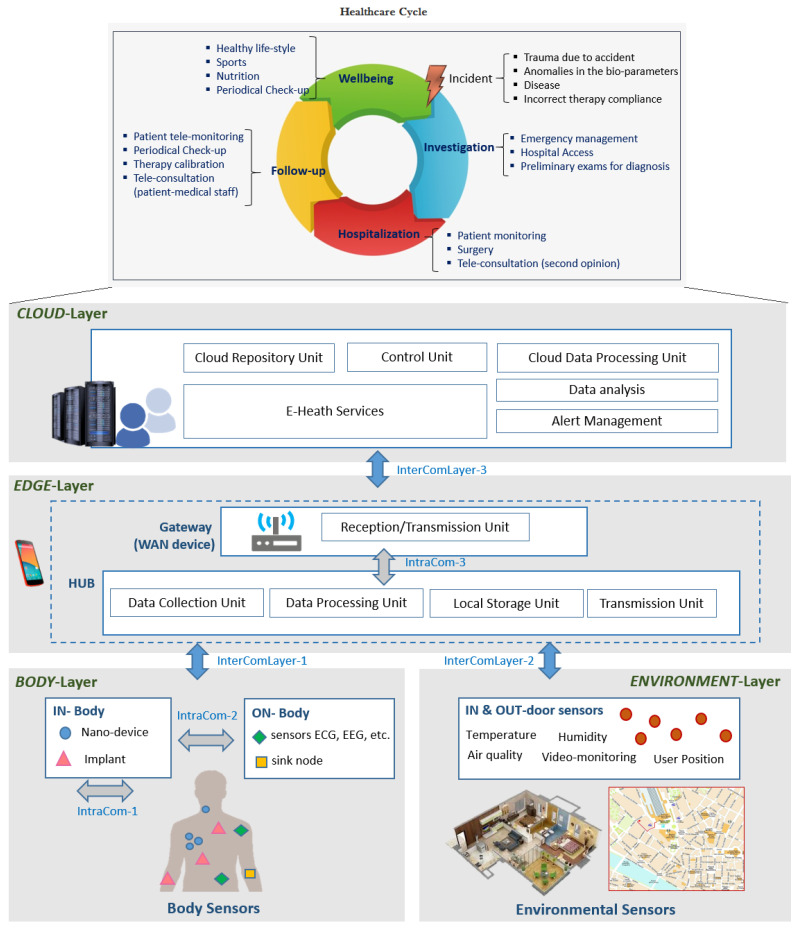
Healthcare Cycle and Reference Network Architecture.

**Figure 3 ijerph-18-07459-f003:**
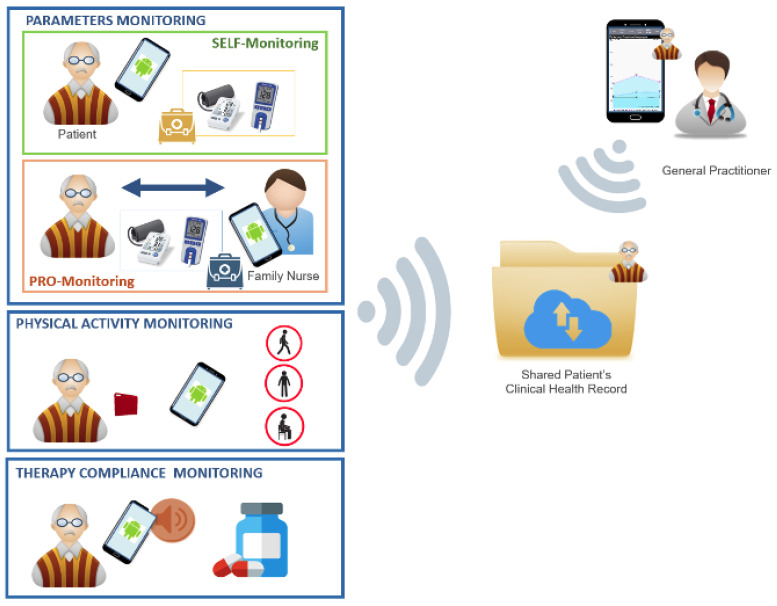
Overview of the System Monitoring Components and Data Sharing.

**Figure 4 ijerph-18-07459-f004:**
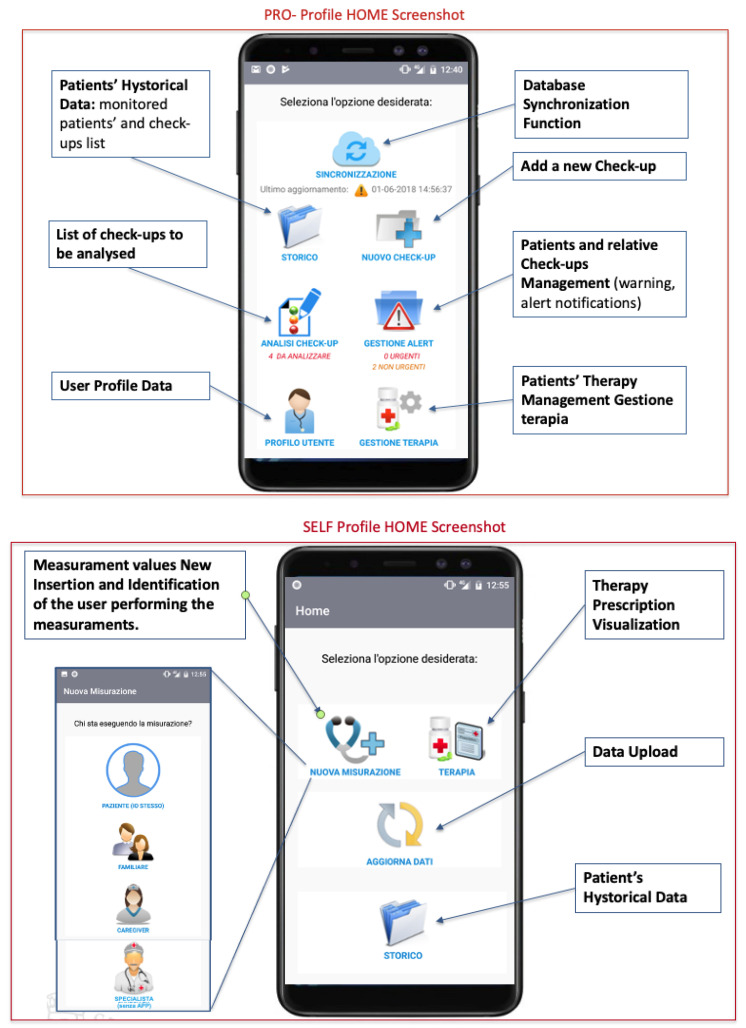
App GUI: Example of screenshots- PRO-Profile and SELF-Profile Home.

**Figure 5 ijerph-18-07459-f005:**
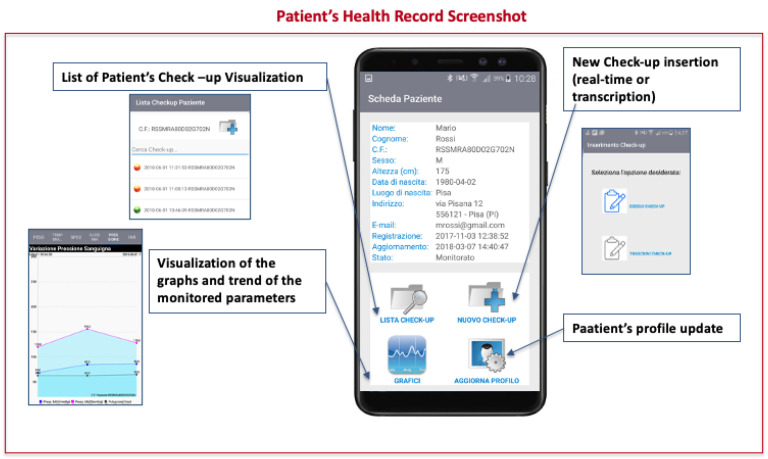
App GUI: Example of screenshots- Patient Health Record.

**Figure 6 ijerph-18-07459-f006:**
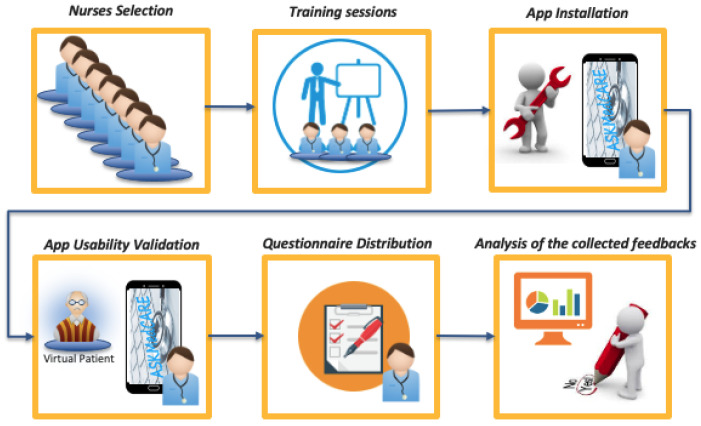
Preliminary Experimental Activity: Stages Overview.

**Figure 7 ijerph-18-07459-f007:**
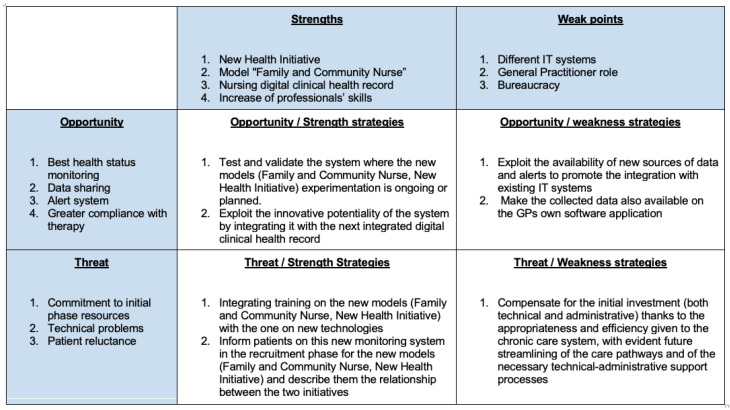
SWOT Analysis.

**Table 1 ijerph-18-07459-t001:** Quality of Experience results.

Investigated Aspect	Insufficient	Sufficient	Fair	Good	Excellent
Nurse ICT skills	-	-	67%	33%	-
App Functionalities with respect to operation conditions	-	-	33%	50%	17%
App Usability in operative contexts	-	-	33%	50%	17%
Approval of the service and the App	-	-	67%	33%	-
System efficiency for the management of chronic diseases	16%	33%	17%	17%	17%

## Data Availability

Not applicable.

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
