# Peer review of "Tele-Monitoring System for Chronic Diseases Management: Requirements and Architecture"

_ijerph, 2021, doi:10.3390/ijerph18147459_

Round 1

Reviewer 1 Report

Thank you for the opportunity to review this paper.  The paper seeks to describe an architecture model for telehealth services within a telemonitoring concept for health and introduction of the family community nurse in the role of patient care using the model.  There is no question that telehealth is gaining in acceptance and innovative approaches since the pandemic.  Telehealth had already become an accepted and appreciated form of care delivery prior to the pandemic, but then accelerated with homebound patients.  There are issues for improvement.  First, I had to read it several times to catch the significance of the paper.  The paper goes through a long bit about the WHO and telehealth recommendations. This doesn't add much to the overall point of the paper.  The point here is the gaining acceptance of telehealth, the model, and evaluation of the nurses. The background could include not only support through the stats on chronic illnesses, but also the technology and nursing. Bring these together with the study.  IRB approval or human subjects approval, the instrument, sample number and description, methods of analysis, results and limitations.  A table would have been useful with the results as these are difficult to follow.   There are a number of variables that could be important with the nurses-years of experience, tech savviness, and so forth.  Although N=10 is small, there could be important bits of information about their perceptions-and not knowing the tool, these are difficult to extract.  General writing of the paper could be strengthened with one idea per paragraph, citations rather than just reference list  number in the text, and decreasing the size of some paragraphs to a manageable size for reading and comprehending.  

Author Response

Reply to Reviewer 1

Comment 1: First, I had to read it several times to catch the significance of the paper.  

Reply to Comment 1: The structure of the paper has been revised and to better highlight the objective of the paper section 1.3  has been updated. 

Comment 2: The paper goes through a long bit about the WHO and telehealth recommendations. This doesn't add much to the overall point of the paper.  

Reply to Comment 2: The introduction aims at providing an overview of both the statistics on chronic diseases and aging and the existing designed and developed telemonitoring solutions  to cope with this ingìcreasing number of patients. As highlighted in the “Our contribution” section, this analysis has been the starting point for the identification of the weakness that needs to be addressed in this context and consequently for defining the requirements of the proposed architecture. The introduction has been revised and reduced to meet the reviewer’s comment, however, according to the authors’ opinion a further reduction of the introduction may not be beneficial for the overall paper.

Comment 3:The point here is the gaining acceptance of telehealth, the model, and evaluation of the nurses. The background could include not only support through the stats on chronic illnesses, but also the technology and nursing. Bring these together with the study. 

Reply to Comment 3: In the Introduction section, the authors focus on the background on chronic diseases because the developed Application, which represents the end-user interface to the system platform, was designed in accordance with the indications of the WHO and the main national and regional guidelines on the management of chronic diseases remotely.

However,  a more general overview on the telehealth systems is provided in section 2.1 while in section 2.2 the main end-users' needs in the management of patients affected by chronic diseases are presented considering the Family and Community Nurse.

Moreover, the presented results refer to the first pilot phase of the project, which aims at involving a small group of end users belonging to the community of nurses for validating the App based on the professional point of view pursuing a collaborative and co-creative approach.

The next and second phase of the experimental activities will aim at validating the app in terms of usability based on the patients’ point of view. They will be involved in receiving feedback that will also be evaluated together with the ones received from the nurses.

The overall paper has been revised in order to better clarify this point. 

Comment 4: IRB approval or human subjects approval, the instrument, sample number and description, methods of analysis, results and limitations.  A table would have been useful with the results as these are difficult to follow.   There are a number of variables that could be important with the nurses-years of experience, tech savviness, and so forth.  

Reply to Comment 4: Since the test phase did not involve real patients, the only approval needed for this kind of experimental activity was the one given by the Director of the Nurse Department, and not by the ethical committee of the health authority. We got the authorization from the Director to go ahead with the experiment with nurses (RN).

In this phase of the experimental activities,  family and community nurses (FCNs)  were involved for the App validation. Each of them only simulated day-by-day the insertion of (fake) check-ups of one or more patients, in order to test the APP functionalities and test the ICT system backend.

The “Experimental activities” section has been fully revised in order to address the reviewer’s comments. Moreover, additional results have been included and,  besides  a revision of the results description, more details of the carried out analysis have been added with the aim to better highlight the experimental activities output. 

Comment 5: Although N=10 is small, there could be important bits of information about their perceptions-and not knowing the tool, these are difficult to extract.  

Reply to Comment 5: The authors would like to underline that this preliminary validation phase of some of the functionalities of the system allows the designer and developers to better understand the weak as well as the strong points of the system. This is in line with the co-design approach that has been followed by the authors. In order to better highlight the output coming from the experimental campaign of this small pilot the “Experimental activities” section has been updated.

Comment 6: General writing of the paper could be strengthened with one idea per paragraph, citations rather than just reference list  number in the text, and decreasing the size of some paragraphs to a manageable size for reading and comprehending.  

Reply to Comment 6: The structure of the paper has been updated together with the length of the subsection to make the article more readable. Moreover the aim of each paragraph has been highlighted.

Regarding the “citations comment” the authors do not understand the reviewer’s suggestion. The references are not just listed, but they are cited in the paper text. 

Reviewer 2 Report

The introduction presents the background and the motivation nicely. However, compared to the level of details given there, the paper misses to describe the actual system and details about the evaluations. 

It is not clear, what features of the general architecture were implemented in the prototype application, nor how it looked (since, usability was part of the questionnaire, some screenshots would be expected to see parts of the UI).

Security and data privacy is not mentioned. This is a very important requirement of every eHealth system. How is it ensured that the medical data is secure?

Describe abbreviations eHealth and mHealth before their first use. Also difference between them. Use same form (not e-health).

Check and unify the capitalization of words. The same word is sometimes written differently throughout the paper (e.g. healthcare system).

Author Response

Reply to Reviewer 2

Comment 1: The introduction presents the background and the motivation nicely. However, compared to the level of details given there, the paper misses to describe the actual system and details about the evaluations. 

Reply to Comment 1: The description of the developed and validated system has been added together with technical details. Moreover the experimental activities section has been updated improving the evaluation presentation both in terms of objectives and results analysis.

Comment 2: It is not clear, what features of the general architecture were implemented in the prototype application, nor how it looked (since, usability was part of the questionnaire, some screenshots would be expected to see parts of the UI).

Reply to Comment 2: The description of the implemented components of the proposed system architecture has been added.  Technical details of the prototype application have been included, together with some GUI screenshots to provide some examples of the user interface that has been one of the components evaluated in the experimental activities.

Comment 3: Security and data privacy is not mentioned. This is a very important requirement of every eHealth system. How is it ensured that the medical data is secure?

Reply to Comment 3: In order to address the security and privacy issues related to the e-health system, a specific section (section 3.4) has been added. 

Comment 4: Describe abbreviations eHealth and mHealth before their first use. Also difference between them. Use same form (not e-health).

Reply to Comment 4: The text has been updated according to the reviewer’s comment. 

Comment 5: Check and unify the capitalization of words. The same word is sometimes written differently throughout the paper (e.g. healthcare system).

Reply to Comment 5:  The text has been revised according to the reviewer’s comment. 

Reviewer 3 Report

This paper aims to present a modular telemedicine system to support medical staff provision of home healthcare assistance to patients affected by chronic diseases. The authors explained an architectural model in line with the Family and Community Nurse (FCN). In terms of empirical analysis, they conducted a preliminary experiment with ten nurses to analyze the usability of an app focused on the patient's parameters monitoring. Preliminary results suggest that the feedback was positive.

Despite the relevance of the content, the paper lacks practical/technical details and methodological rigor (especially considering that we are dealing with a journal publication). More details must be provided regarding the ICT architecture model, especially in light of the component submitted for experimental evaluation. I understand that the work is under development, but almost nothing about the app to monitor the patient's parameters is provided. For example: How do the system requirements were collected and specified, including the service platform? What were the references for the network architecture? How does the proposed ICT model differentiates/contributes when compared to state of the art? How does the app work in practice? How does the app cope with scalability, portability, low cost, and user-friendly since they were found as key factors for the adoption?

The authors also do not provide sufficient information regarding the experimental design, including the participant characterization (experience, profile, etc), the training session, the evaluation questionnaire, etc. In addition to mentioning what was done, it is quite essential to describe how it was accomplished.

Finally, the results and analysis were oversimplified. For instance, what evidence grounds the social impact factors identified? How do the authors claim that their proposed system is effective (line 404) if only one of its components was evaluated? Why do the authors opt for the Mean Opinion Score scale instead of other widely adopted usability scales? Why did the experimental design approach only the nurses' perception? What were the limitations faced by the authors, and how were they mitigated? What were the derived guidelines for the implementation of the whole system?

IMHO, this series of unanswered questions threatens the study's validity and, consequently, requires a significant paper revision.

Best regards.

Author Response

Reply to Reviewer 3

Comment 1: Despite the relevance of the content, the paper lacks practical/technical details and methodological rigor (especially considering that we are dealing with a journal publication). More details must be provided regarding the ICT architecture model, especially in light of the component submitted for experimental evaluation. I understand that the work is under development, but almost nothing about the app to monitor the patient's parameters is provided. 

Reply to Comment 1: The paper has been updated according to the reviewer comment. The structure has been modified and additional information regarding the proposed system and the developed and tested components has been included, providing a deeper overview of the system in terms of technical aspects. For addressing all the points highlighted by the reviewer, please refer to the “Reply to Comment 2-6”

Comment 2: For example: How do the system requirements were collected and specified, including the service platform? 

Reply to Comment 2: The system requirements derive from the end user requirements. In detail, the end-user requirements for patient monitoring (self- and professional monitoring) have been defined,  focusing on the main application context of the proposed solution “Tele-monitoring services for patient assistance and follow up at home”. These requirements have been identified thanks to a deep collaboration between the Department of Information Engineering and Local Health Unit Toscana Centro, in particular the Nursing and Midwifery Department. This collaboration has been fundamental for the design and the development of the tested components of the proposed architecture. In detail, it allowed the definition of the App functionalities and interfaces, and consequently the service requirements, in compliance with the medical professional end-users needs, also taking into account their organizational models and operative contexts. 

The text of the paper has been updated including the previous considerations, in order to provide the reader with a better understanding of  the design and co-design approach followed from the very beginning of the system definition.

Comment 3: What were the references for the network architecture? 

Reply to Comment 3: The authors would like to thank the reviewer for his/her help in improving the quality of the paper. Unfortunately, the authors did not understand the meaning of this specific comment. In particular, we cannot understand if the reviewer wanted to highlight that the references for the network architecture were missing in the first version of the manuscript, or something different from this. The network architecture was designed by the authors for the specific experimentation carried out by the volunteer nurses.   

Comment 4: How does the proposed ICT model differentiates/contributes when compared to state of the art? 

Reply to Comment 4: As highlighted in the “background” and in the “our contribution” sections the existing telemonitoring solution in the eHealth context mainly focused on the provision of a specific service. The aim of this paper is to provide a general tele-monitoring system that can be easily adapted to most of the health services. The system capability of  gathering data coming from different sources including both the human body and the surrounding environment allows the definition of advanced services based on the correlation of multiple parameters, providing a more accurate real time evaluation of the patient's health status enabling a multi-purposes data analysis.

Our contribution differs from the state of the art mainly for the co-creation and co-design approach among engineers and medical staff followed from the very beginning of this study.  This allows to define an integrated and personalized real-time tele-monitoring solution focused on the real operative conditions and user requirements, including data validation procedures and alert management. This multidisciplinary co-design approach is the key element of the proposed system.

The text of the paper has been updated including the previous consideration.

Comment 5: How does the app work in practice? 

Reply to Comment 5: The description of how the App work in practice has been added in the text. In particular a new section has been added (section 4) to describe the user interface for interacting with the system platform. The App details have been provided together with practical examples on its functioning, including the operative use case and the relative information flow. 

Comment 6: How does the app cope with scalability, portability, low cost, and user-friendly since they were found as key factors for the adoption?

Reply to Comment 6: The App has been designed following a modular approach that allows an easy integration of future functionalities. This flexible solution also allows scalability in terms of user typology and specific application contexts. The main idea is to provide an App whose mail functionalities can be activated based on the patients  clinical and management needs. 

The App GUI is intuitive and user friendly. Some screenshots of the App developed and tested functionalities have been integrated in the text together with a more detailed analysis of the obtained results. 

Comment 7: The authors also do not provide sufficient information regarding the experimental design, including the participant characterization (experience, profile, etc), the training session, the evaluation questionnaire, etc. In addition to mentioning what was done, it is quite essential to describe how it was accomplished.

Reply to Comment 7: The Experimental activities and the results sections have been updated and reorganized in order to provide the requested information.  Experimental activities planning, execution details and achieved results are described.  

Comment 8: Finally, the results and analysis were oversimplified. For instance, what evidence grounds the social impact factors identified? 

Reply to Comment 8: The results analysis has been deeply revised. It has been organized in different subsections with the aim to address all the reviewer’s comment.

In detail: the results have been better presented (Table 1 has been added).

Additionally, the aim of the collected feedback has been highlighted and some of the questions included in the questionnaire submitted to the nurses (RNs) involved in the preliminary validation phase  have been reported.

Moreover the expected impact has been better described and a SWOT analysis has been added with the aim to give an overview of the strengths, weaknesses, opportunities, and threats related to this  study.

Comment 9: How do the authors claim that their proposed system is effective (line 404) if only one of its components was evaluated? 

Reply to Comment 9: The authors agree with the reviewer comment. The term “effective” can refer to the tested components. The text has been updated.

Comment 10: Why do the authors opt for the Mean Opinion Score scale instead of other widely adopted usability scales? 

Reply to Comment 10: In order to clarify this point the authors have deeply revised the Experimental activities and Results sections.

In particular the Experimental activity plan has been described, highlighting that the reported results refer to the first phase of the experimental campaign which involves a small number of  RNs and which is preliminary to the second phase which will involve both RNs and patients in a larger scale.

Therefore, in this paper  the results of a pilot phase of a larger study is presented. In this first phase, a preliminary ad-hoc questionnaire has been defined by the Dept. of Information Engineering of the University of Florence to evaluate the usability of the product. The next larger phase (second) will involve the introduction of the app on an experimental basis in clinical practice, including both nurses and  patients of a selected area of the Local Health Unit Tuscany Centre.

To achieve the objectives of the experimentation and the evaluation of the QoE (Quality of Experience), the following data were collected:

  •   approval rating
  •   index of functionality
  •   usability index
  •   acceptability index

These indices were measured according to the MOS (Mean Opinion Score) scale and were the output of a questionnaire submitted to the nurses involved in the pilot phase.

The analysis of the data obtained allows the authors to improve the App.

These considerations have been included in Sections 5 and 6.

Comment 11: Why did the experimental design approach only the nurses' perception? 

Reply to Comment 11: The choice of this pilot phase is to develop the tool in close collaboration with the end user in a co-design perspective that will be further refined after the second experimental phase in which patients will also test the app and express their opinion.

Once the app will be updated based on the received feedback of the second phase of the experimental campaign, its use can be further extended to the entire territory of the Local Health Unit Toscana Centro (which covers an area of 5000 square kilometers and 1,500,000 citizens, it has 13 hospitals, 220 territorial care centers and including both urban areas with high population density and sparsely populated rural and mountain areas).

In fact, from this experience it will be possible to develop solid guidelines for the adoption of a new patient-nurse monitoring systems dedicated to chronic patients to be spread throughout the Tuscany region.

These considerations have been included in Sections 5 and 6.

Comment 12: What were the limitations faced by the authors, and how were they mitigated? 

Reply to Comment 12: The first faced limit is the involvement of as many stakeholders as possible in the co-creation and co-design phase of the monitoring system. We mitigate this limit by proposing a multiphase trial at the end of which the monitoring system will be updated and modified based on the feedback  received from the end users: both nurses and patients.

Another limit could concern the ICT skills necessary for the use of the tool: this is controlled precisely because the monitoring system is built as user friendly as possible taking into account that the final patients are elderly people who are not familiar with new technologies. 

Comment 13: What were the derived guidelines for the implementation of the whole system?

Reply to Comment 13: The App design followed the current guidelines for the implementation of the telemonitoring systems indicated also by the WHO and by the national and regional health plans in addition to the indications provided by the scientific societies for chronic diseases.

Guidelines for the application of this monitoring model will be defined and disseminated, initially at the regional level, after the second experimentation phase, which will involve both  patients and all local family nurses. These guidelines will be based on the experience of a large number of patients and nurses involved in the validation of the proposed App.

Section 6.3 has been added in the paper to address this comment. 

Comment 14: IMHO, this series of unanswered questions threatens the study's validity and, consequently, requires a significant paper revision.

Reply to Comment 14: Thank you for all the useful suggestions which gave us the opportunity to improve the manuscript. The authors have revised the paper, including the answers to all the reviewer’s comments.  According to the authors opinion, the overall quality of the paper has been improved, allowing the reader to better understand the study’s validity.

Round 2

Reviewer 2 Report

Thanks a lot for the various improvements of the paper, especially the more detailed description about the implementation and usability. 

Author Response

Reviewer 2

Comment 1. Thanks a lot for the various improvements of the paper, especially the more detailed description about the implementation and usability.

Reply to comment 1. All the authors would like to thank very much the reviewer for the very useful comments and suggestions which gave us the opportunity to improve the overall quality of the revised manuscript. 

Reviewer 3 Report

I appreciate that the authors have demonstrated special attention in answering all the questions, and, consequently, considerable improvements were made. 

I have two minor suggestions that the authors could consider in the final version of the manuscript. Could the authors provide more information about the co-creation and co-design process performed to derive the end-user requirements (not only in terms of usability validation)? This issue is particularly important considering the “Reply to comment 4” in which the authors highlighted the value of this stage. Regarding the network architecture, the authors could specify and clarify what components are based on other works (for example, the healthcare cycle).

I wish the authors all the best with this project that is especially relevant given the critical health challenges that we face as a society.

Author Response

Reviewer 3

I appreciate that the authors have demonstrated special attention in answering all the questions, and, consequently, considerable improvements were made. 

I wish the authors all the best with this project that is especially relevant given the critical health challenges that we face as a society.

All the authors would like to thank the reviewer very much for the very useful comments and suggestions which gave us the opportunity to improve the overall quality of the revised manuscript.

I have two minor suggestions that the authors could consider in the final version of the manuscript. 

Comment 1. Could the authors provide more information about the co-creation and co-design process performed to derive the end-user requirements (not only in terms of usability validation)? This issue is particularly important considering the “Reply to comment 4” in which the authors highlighted the value of this stage. 

Reply to comment 1. The co-creation and co-design approach followed by the authors relies on their awareness that an effective ICT solution, which includes Human-Computer Interaction (especially in the healthcare sector) should be based on the adoption of a participatory design paradigm,  that allows to satisfy the end-users’ needs  and, consequently, improve their quality of experience. The participatory design (PD)  main principles are deep engagement, interdisciplinary, individuality, and practicality. 

The proposed system comes from an interdisciplinary investigation based on the PD approach since the very beginning of the study. The deep collaboration between engineers and end-users (e.g. registered nurses, medical staff) has allowed the identification of the existing needs.

This has enabled the definition of real use cases taken as a reference to show the benefit of the proposed system with respect to the already existing and sometimes adopted technologies. Moreover it has allowed the identification of the main potential challenges that need to be taken into account both in the development and validation phase, but also to make the system be accepted and therefore adopted by the end-users. 

The previous considerations have been integrated in the text.

Comment 2. Regarding the network architecture, the authors could specify and clarify what components are based on other works (for example, the healthcare cycle). 

Reply to comment 2. Regarding the proposed network architecture, it is worth highlighting that the authors from the Department of Information Engineering are part of the ETSI SmartBAN Working Group and actively participate in the WG activities. The proposed architecture is based on the high level concept of the architecture defined within the WG activities [1]. 

[1] M. Hämäläinen, L. Mucchi, et al., "ETSI SmartBAN Architecture: The Global Vision for Smart Body Area Networks," IEEE Access, vol. 8, pp. 150611-150625, 2020. 

On the other hand, the healthcare cycle (depicted in Figure 2) derives from: the revision of the state of the art on the healthcare available services, the analysis of the main user requirements and the author's experience in the healthcare sector.

The previous considerations have been integrated in the text.